evolution, genomics

genome evolution, macrosynteny conservation, microsynteny conservation, developmental gene clusters, Lepidoptera, Diptera

**Author for correspondence:**
José M. Ranz
e-mail: jranz@uci.edu

†Present address: Wellcome Trust Sanger Institute, Wellcome Trust Genome Campus, Hinxton, UK.

‡Present address: Institute of Evolutionary Biology, School of Biological Sciences, University of Edinburgh, Edinburgh, UK.

# Multiscale analysis of the randomization limits of the chromosomal gene organization between Lepidoptera and Diptera

José M. Ranz[1], Pablo M. González[2,†], Ryan N. Su[1], Sarah J. Bedford[1], Cei Abreu-Goodger[2,‡] and Therese Markow[2,3]

[1]Department of Ecology and Evolutionary Biology, University of California Irvine, Irvine CA 92647, USA
[2]Unidad de Genómica Avanzada (Langebio), CINVESTAV, Irapuato GTO 36824, México
[3]Section of Cell and Developmental Biology, Division of Biological Sciences, University of California San Diego, La Jolla, CA 92093, USA

JMR, 0000-0003-3585-3129

How chromosome gene organization and gene content evolve among distantly related and structurally malleable genomes remains unresolved. This is particularly the case when considering different insect orders. We have compared the highly contiguous genome assemblies of the lepidopteran *Danaus plexippus* and the dipteran *Drosophila melanogaster*, which shared a common ancestor around 290 Ma. The gene content of 23 out of 30 *D. plexippus* chromosomes was significantly associated with one or two of the six chromosomal elements of the *Drosophila* genome, denoting common ancestry. Despite the phylogenetic distance, 9.6% of the 1-to-1 orthologues still reside within the same ancestral genome neighbourhood. Furthermore, the comparison *D. plexippus–Bombyx mori* indicated that the rates of chromosome repatterning are lower in Lepidoptera than in Diptera, although still within the same order of magnitude. Concordantly, 14 developmental gene clusters showed a higher tendency to retain full or partial clustering in *D. plexippus*, further supporting that the physical association between the SuperHox and NK clusters existed in the ancestral bilaterian. Our results illuminate the scope and limits of the evolution of the gene organization and content of the ancestral chromosomes to the Lepidoptera and Diptera while helping reconstruct portions of the genome in their most recent common ancestor.

## 1. Introduction

With the exception of neighbouring genes featuring coordinated gene expression through shared or long-range enhancers [1–3], gene order organization among distantly related metazoans is thought to be quasi-random as a result of chromosome structural mutations [4]. Some insect orders are particularly well suited to test for the limits to this quasi-randomization as they possess some of the most structurally dynamic genomes among eukaryotes [5–7]. However, how chromosome gene organization and content have evolved among these fast-evolving insect orders remains unresolved primarily because of the lack of high-contiguity genome assemblies [8], the absence or incompleteness of species gene sets [9], the difficulty to establish reliable orthologues relationships in the absence of phylogenetic gene trees [10], or any combination of these factors. Consequently, studies on the evolution of their chromosome gene organization and content have been primarily restricted to a few reference species and their closest relatives within the same genus or order [5,6,11]. The employment of long-sequencing read technologies coupled with advanced

assembly scaffolding tools [12–14], improved *ab initio* prediction tools along with the implementation of RNA-seq for enhanced gene annotation [15] and the development of methods for phylogenetic orthology inference [10] allow us to examine this question. Thus, we now are well poised to revisit, or address for the first time, crucial aspects of the evolution of the gene organization at the chromosomal level between insect orders.

Lepidoptera and Diptera are two of the four most species-rich orders within the insects [16], with their most recent common ancestor dating back approximately 290 Ma [17]. Crucially, both Lepidoptera and Diptera chromosomes are characterized by high levels of co-localization of genes within equivalent chromosomes between species (i.e. macrosynteny) but low levels of conservation of gene organization at a fine scale (i.e. microsynteny), and with chromosomal fissions/fusions and paracentric inversions reshaping the karyotype and chromosome architecture within these species orders [5,6,18–20]. Isolated efforts have attempted to clarify the origin of the gene content of the Lepidoptera and Diptera heterochromosomes Z and X, respectively [21], and to examine the dynamics of change in gene configuration of a limited number of genomic regions, including the clusters of developmental genes Hox and Wnt [5,22–24], the *Osiris* multigene family [25], and 15 random regions representing approximately 2% of the genome of two noctuid moths [18]. Therefore, no effort has been performed so far to (i) comprehensively investigate the relationship between the chromosomes between the Lepidoptera and the Diptera, (ii) determine the extent to which gene order randomization has taken place between the species orders, and (iii) reconstruct, even partially, the chromosome gene organization in their ancestor. To fill these gaps in knowledge, we used a recently generated high contiguity assembly and gene annotation for *D. plexippus*, as a representative of the Lepidoptera, as well as the inferred orthologous relationships between this species and *D. melanogaster* [26]. In addition, we also gauged the scope of gene reshuffling within the Lepidoptera by comparing the gene organization between *D. plexippus* and *B. mori*, a representative of the moths, thus obtaining a reliable estimate of breakpoint occurrence that can be fairly compared to those obtained in the genus *Drosophila*. Our results demonstrate the potential of high-quality genomic resources in uncovering signatures of macro- and microsynteny, helping ultimately to reconstruct the genome in the ancestor of structurally dynamic genomes of distantly related species.

## 2. Methods

### (a) Orthologous gene sets

1-to-1 orthologues between *D. plexippus* and *B. mori*, and between the former and *D. melanogaster*, were previously delineated using OrthoFinder v. 2.2.6 [27] under the settings –S diamond –M msa, using protein models retrieved from either NCBI or lepBase [26], and ultimately linking the orthologues to their corresponding genomic coordinates in each species. Ninety-five orthologues potentially involved in interchromosomal gene transpositions between *D. plexippus* and *B. mori* [26] were not considered in downstream analyses as they are not part of large-scale chromosomal rearrangements. Chromosomal locations of 1-to-1 orthologues were recorded and tabulated for both comparisons.

### (b) Test to the bias in shared chromosomal gene content at the interspecific level

To determine whether the contemporary chromosomal distribution of 1-to-1 orthologues between *D. plexippus* and *D. melanogaster* reflects to some degree that in their common ancestor, we performed 10 000 permutations of the chromosomal location of such orthologues. The observed chromosomal distribution was compared against the 10 000 generated distributions. The number of orthologues landmarks located in particular *D. plexippus* and *D. melanogaster* chromosomes was recorded. The *p*-values obtained represent the fraction of the simulated distributions in which the number of 1-to-1 orthologues being located in particular *D. plexippus* and *D. melanogaster* chromosomes was equal or higher than that observed. The breadth of gene content association of a given *D. plexippus* chromosome relative to the *D. melanogaster* chromosomes was calculated using the $\tau$ index [28], which takes values from 0 to 1, and it is defined as

$$\tau = \frac{\sum_{i=1}^{n}(1 - x_i)}{N - 1},$$

where $N$ is the number of chromosomal elements in *D. melanogaster* (*X*, *2L*, *2R*, *3L*, *3R* and *4*) and $x_i$ is the number of 1-to-1 orthologues located in each *D. melanogaster* chromosome normalized by the maximum number. In this case, $\tau$ measures the degree of location specificity of 1-to-1 orthologues between the chromosomes of the two species. The higher the value, the larger the fraction of the orthologues harboured by a given chromosome of *D. plexippus* that reside in a particular *Drosophila* chromosomal element. To avoid any distorting effect associated with the comparably limited number of orthologues located on the dot-like chromosome *4* of *D. melanogaster* in relation to the rest, we repeated the analysis excluding this chromosome.

### (c) Fine-scale conservation of gene organization

Global patterns of microsynteny conservation were evaluated using positional information of 1-to-1 orthologues between *D. plexippus* and *B. mori*, and between the former and *D. melanogaster*. Between *D. plexippus* and *B. mori*, we demarcated microsynteny blocks based on conservation of adjacency, not orientation, of at least two 1-to-1 orthologues. In addition to being adjacent in both species, the distance between any two given orthologues was restricted to no more than 0.1% of the assembly size in both species (i.e. 250 kb in *D. plexippus* and 460 kb in *B. mori*). In the case of microsynteny blocks harbouring three 1-to-1 orthologues, the precise gene order did not need to be identical. Presence of 1-to-1 orthologues associated with either inter- and intra-chromosomal gene transpositions, which are known to be very infrequent [5,26], was not considered to disrupt microsynteny. Likewise, putative alterations of local gene order due to differential gene configurations between the species (e.g. one of the genes being nested into another in one species but overlapping with an adjacent gene in the other) were not considered to disrupt microsynteny as they could be annotation errors. Two estimates of gene order disruption were obtained. The first, a minimum estimate, corresponds to the number of microsynteny blocks minus the number of chromosomes as for each chromosome the number of breakpoints equals the number of microsynteny blocks minus one. The second estimate was obtained with the GRIMM-Synteny program v. 2.01 [29], using the order and orientation of the demarcated microsynteny blocks as input. This program assumes maximum parsimony while accounting for the phenomenon of breakpoint reuse [6]. The number of breakpoints estimated in these two ways, divided by two, corresponds to the inferred number of inversions fixed during the evolution of the lineages that lead

to *D. plexippus* and *B. mori*. Furthermore, in the case of the comparison between *D. plexippus* and *D. melanogaster*, we applied the same rationale including the criterion of 0.1% maximum distance between 1-to-1 orthologues in order to consider them part of the same microsynteny block. Nevertheless, in this interspecific analysis, as it is known that most genes in *D. melanogaster* reside in the euchromatin, we required that the distance be proportional to the size of the euchromatin (i.e. 120 kb).

For the comparative analysis of the gene organization of clusters of developmental genes, we used positional information from 1-to-1 orthologues in *D. plexippus* as well as TBLASTN reciprocal best hit searches between *D. melanogaster* and *D. plexippus* [30]. Homology searches were done by running in-home shell scripts in the UCI High Performance Computer cluster against the recently generated genome assembly DpMex_v1 (scaffold N50 = 8.16 Mb; complete BUSCOs = 98% [26]) of *D. plexippus*, and by using the assemblies, gene annotations and BLAST tools available in Ensembl (http://ensemblgen omes.org/) for *B. mori* (ASM15162v1) and *D. melanogaster* (BDGP6.28). In this analysis, microsynteny conservation was considered to be also distance-dependent (0.1% criterion, i.e. ≤250 kb in *D. plexippus* and ≤120 kb in *D. melanogaster*) between adjacent relevant landmarks, and preserved even in the presence of intervening genes. Clustal Omega was used to visually inspect protein sequence alignments and resolve cases of potential gene model fragmentation [31]. When confirming local gene duplications and deletions were necessary, we extracted genome sequences from genome assemblies and performed local alignments using PipMaker [32].

## (d) Statistical analysis

Statistical analyses including data permutations were performed using built-in functions in R [33]. Individual parameters and statistically significant results are indicated in the text.

# 3. Results

## (a) Uncovering chromosome ancestry between *D. plexippus* and *D. melanogaster*

We investigated the degree of chromosome-level synteny between *D. plexippus* (chromosome number = 30) and *D. melanogaster* (chromosome number = 4) using positional information from 5108 1-to-1 orthologues [26]. Lepidoptera possess holocentric chromosomes and their ancestral chromosome number is thought to be 31 [34]. This number is reduced to 30 in *D. plexippus* as a result of a fusion between the ancestral heterochromosome *Z* and an autosome, pre-dating the radiation of the genus *Danaus* [35,36]. In the Diptera, the ancestral karyotype is thought to be $n = 6$ [11,37], with the different ancestral chromosomal elements also participating in different fusion events [7,38]. The *D. melanogaster* karyotype includes four chromosomes that correspond to the six commonly referred to as Muller's chromosomal elements [39]: the telocentric chromosome *X*, which corresponds to Muller's element A; the metacentric chromosome *2*, which resulted from the fusion between Muller's elements B and C; the metacentric chromosome *3*, which is also a by-product of a fusion between Muller's elements D and E; and the small telocentric dot-like chromosome *4*, which corresponds to Muller's element F. As the karyotypes of the outgroup insect orders [17] Coleoptera and Hymenoptera are intermediate between those of Lepidoptera and Diptera (electronic supplementary material, table S1), it seems plausible that a differential set

of fusions (prevailing in the Diptera lineage) and fissions (prevailing in the Lepidoptera lineage) had involved the chromosomes ancestral to Lepidoptera and Diptera, resulting in the karyotype disparity between *D. plexippus* and *D. melanogaster*. Assuming a limited exchange of gene content among ancestral chromosomal elements mainly via non-Robertsonian translocations and pericentric inversions, which, with a few exceptions, are thought to be uncommon based on the species compared within both orders [5,6,18,19,40,41], a full randomization of such gene content during the evolution of the Lepidoptera and Diptera lineages should not be expected. Nevertheless, a precise description and quantification of the levels of the retained common ancestry at the chromosomal level is still lacking.

We sought patterns of common ancestry at the chromosomal level using positional information from 1-to-1 orthologues between *D. plexippus* and *D. melanogaster*. The median number of 1-to-1 orthologues per *D. plexippus* chromosome was 169 (mean ± s.d., 158.5 ± 78.0; CV = 49.2%). All but chromosome *29* harbour at least one landmark, with chromosome *26* and chromosome *3* featuring the second lowest (60) and maximum (309) number of orthologues landmarks, respectively (electronic supplementary material, table S2). We tested for preferential association of the gene content between particular *D. plexippus* chromosomes and those of *D. melanogaster* (Methods), which denote fractions of the contemporary chromosomes in the two species that derive from the same portion of the chromosomes in the ancestor to the two species orders [21]. We found clearly recognizable patterns of shared common ancestry between the chromosomes of *D. plexippus* and *D. melanogaster* (figure 1a), helping establish unambiguous relationships at the chromosomal level between both species. Specifically, for 23 of the 29 (30 as the anc-Z and neo-Z were considered separately) assessable chromosomes of *D. plexippus*, we found that their gene content was preferentially associated with one or two of the *D. melanogaster* chromosomal elements (at $p < 2.8 \times 10^{-4}$; Monte Carlo simulations).

To further quantify the specificity of association between the gene contents of given chromosomes of *D. plexippus* and *D. melanogaster*, we repurposed the tau index ($\tau$), a measure of expression specificity [28]. *D. plexippus* chromosomes *5*, *18* and *22* showed highly specific associations (i.e. $\tau$ greater than 0.85) with Muller's element E, while chromosome *27* did so with Muller's element C (figure 1b). Interestingly, chromosomes *5*, *22* and *27* harbour most of constituent genes reported to be part of developmental gene classes Hox, NK, and SuperHox, respectively (see below). Furthermore, we also uncovered the tight relationship between the *D. plexippus* autosome *23* and the dot-like Muller's element F of *D. melanogaster*, which in brachyceran dipterans and the outgroup order of the Blattodea corresponds to the *X* chromosome [43,44].

Our results also exposed the independent origin of the heterochromosome ($Z/X$) between *D. plexippus* and *D. melanogaster*, as well as the differential chromosomal substrate of the two arms of the *Z* chromosome, extending previous inferences based on a similar approach between *D. melanogaster* and *B. mori* that used approximately 70% fewer 1-to-1 orthologues [21]. Specifically, while the anc-Z of *D. plexippus*—chromosome *1* of *B. mori*—, along with autosomes *11*, *12* and *20*, are enriched for orthologues located on Muller's element D (3L), the neo-Z (chromosome *16* of *B. mori* [26]), along with autosomes *3*, *7*, *9*, *15* and *19*, show a tight

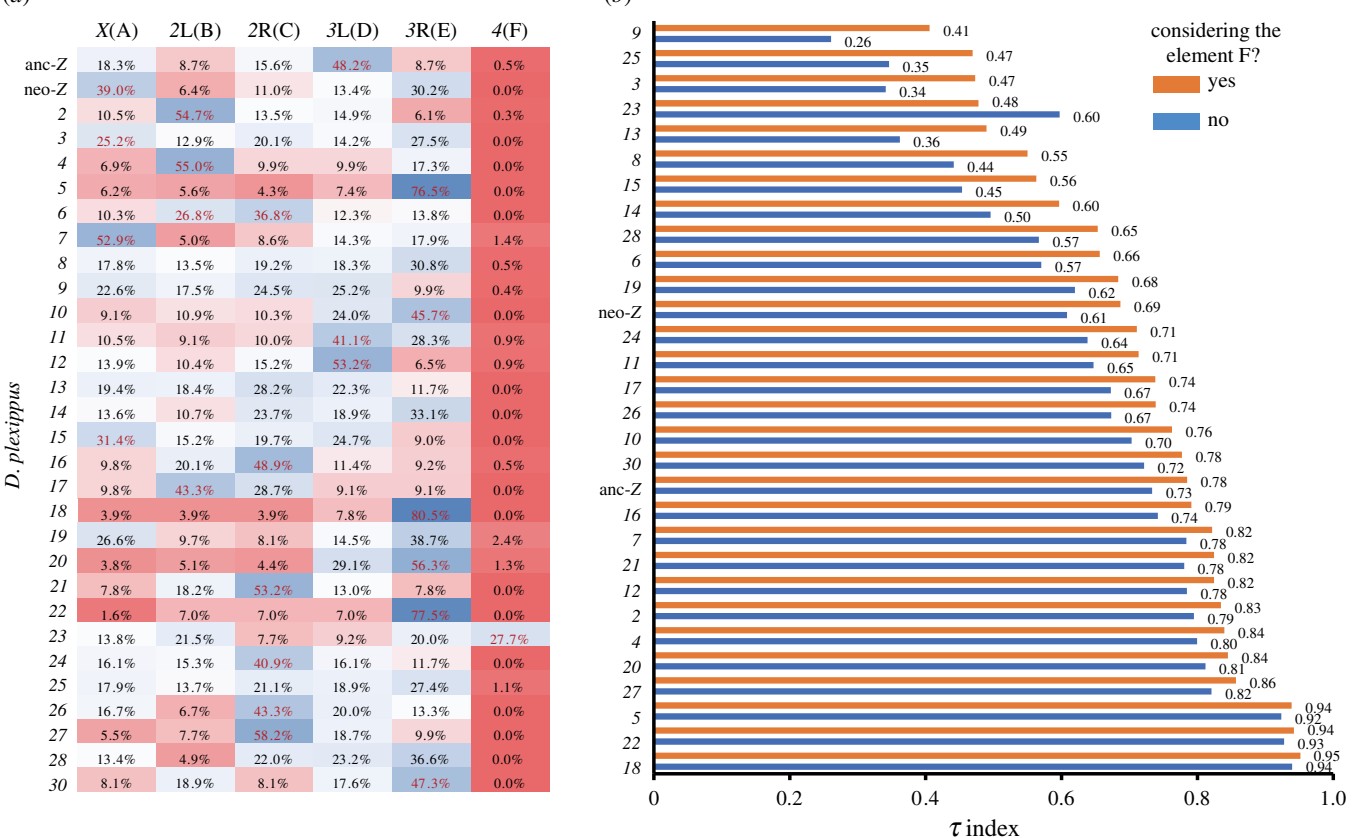

**Figure 1.** Synteny relationships between the chromosomes of *D. plexippus* and *D. melanogaster*. (*a*) Heatmap showing the differential percentage of 4913 1-to-1 orthologues from each chromosome of *D. plexippus* that resides across the ancestral chromosomal elements in the genus *Drosophila* (top). These elements are referred to as A–F [39]. Blue, high percentages; red, low percentages. When the percentage observed is equal or higher than expected by chance alone at $p < 0.00028$ based on Monte Carlo simulations is indicated in magenta. Twenty-three of the 29 assessable chromosomes (chromosome *29* harbours no detectable 1-to-1 orthologue under the parameters used) showed a significant association in gene content with particular chromosomal elements of *Drosophila*. Essentially, the same conclusions are reached based on the analysis of the Pearson's standardized residuals [42] (not shown). For 14 of those chromosomes, the significant association results into more than twofold difference between the largest contribution and the second largest contribution to a different *Drosophila* chromosomal element. (*b*) Chart ranking the chromosomes of *D. plexippus* from lower to higher tau ($\tau$) index [28]. The higher the value (0, min; 1, max), the tighter is the association between the gene content of a particular *D. plexippus* chromosome and a given *Drosophila* chromosomal element (i.e. the larger the fraction of the orthologues harboured by a given chromosome of *D. plexippus* that reside in the same *Drosophila* chromosomal element). Given the much smaller size of Muller's element F relative to the remainder of the Muller's elements, the tau index was calculated including and omitting such chromosomal element. (Online version in colour.)

association with Muller's element A ($X$) of *D. melanogaster* (electronic supplementary material, figure S1).

## (b) Incomplete full gene order randomization between Lepidoptera and Diptera

Given the presumed exceptional properties that the Lepidoptera and the Diptera offer (i.e. high rates of structural rearrangement [5,6] coupled with large phylogenetic distance between these species orders), microsynteny conservation is more likely to reflect functional constraints associated with complex regulation of gene expression than mere phylogenetic inertia [1–3,45,46] (i.e. conservation due to the serendipitous non-occurrence of breakpoints of structural rearrangements). Using positional information from 1-to-1 orthologues previously delineated [26], we sought evidence of microsynteny conservation based on physical distance among neighbouring orthologues (≤0.1% of the total assembly size in both species) at two phylogenetic scales: within the Lepidoptera, as previous analyses in this order used early genome assemblies more affected by fragmentation [5] or

focused on small genomic fractions [18]; and between the Lepidoptera and the Diptera, as previous analyses relied on lower numbers of orthologues [21].

First, we compared the patterns of gene order contiguity between *D. plexippus* and *B. mori*, a moth species whose most recent common ancestor with the butterflies dates back to approximately 100 Ma [47], using information from 7145 1-to-1 orthologues [26], which results in one orthologues landmark per 33–64 kb of the genome of the species, respectively. The genome of *D. plexippus* and *B. mori* can be envisioned as a collection of 647 microsynteny blocks containing at least two 1-to-1 orthologues, including in total 7007 (98.1%) of all 1-to-1 orthologues considered, with approximately 52% of these blocks harbouring ≤5 1-to-1 orthologues (mean ± s.d., 10.84 ± 13.99; CV = 129%). These microsynteny blocks would have been reshuffled within the contemporary versions of ancestral chromosomes by 308–337 inversions (electronic supplementary material, note S1), disrupting the overall collinearity along the chromosomes of these species to different extents, although in no case involving large-scale exchanges of gene content between different chromosomes, including

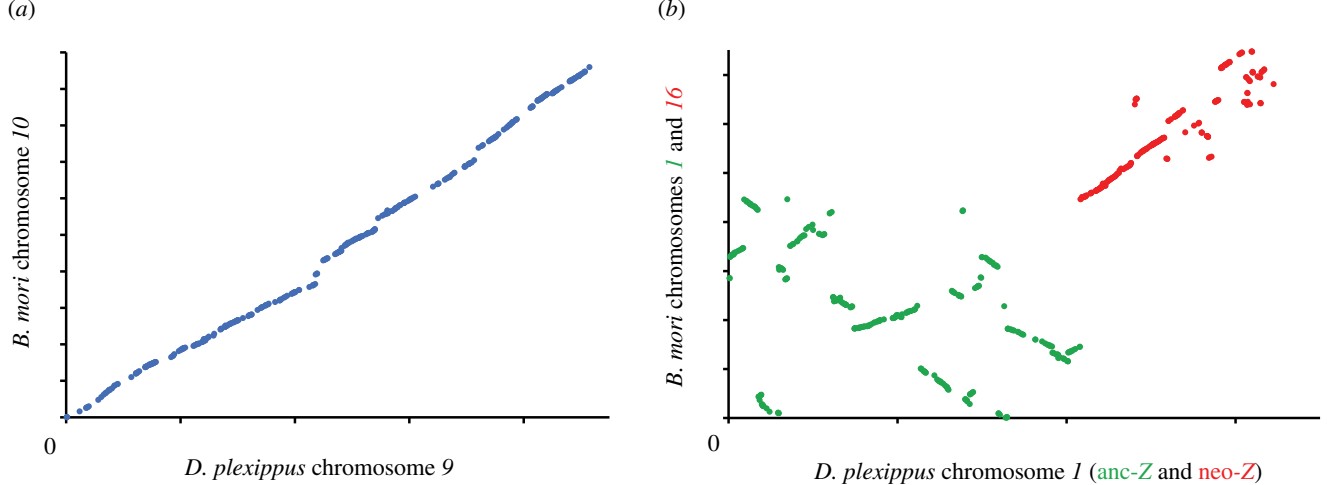

**Figure 2.** Chromosome remodelling between *D. plexippus* and *B. mori*. The dot plots are based on positional information from 1-to-1 orthologues between the two species. While some ancestral chromosomes have barely accommodated major large-scale structural changes, therefore preserving general gene order collinearity as denoted by a well-defined diagonal (*a*), others have undergone a much more profound reorganization, as denoted by a severely disrupted diagonal with multiple microsynteny blocks in different orientation and location along orthologues chromosomes (*b*). Thus, chromosome *9* in *D. plexippus* (*10* in *B. mori*) exhibits one of the lowest breakpoint densities per Mb. By contrast, both chromosomal components of heterochromosome *Z* in *D. plexippus* exhibit some of the highest breakpoint densities per Mb (electronic supplementary material, table S3). (Online version in colour.)

those that are composites of two ancestral chromosomes due to fusion events (figure 2). Rates of gene order repatterning across chromosomes varied substantially (electronic supplementary material, table S3), being overall 3.21–6.69 lower than those reported in the genus *Drosophila*, but still within the same order of magnitude, thus confirming that both species orders are characterized by high rates of remodelling of gene organization (electronic supplementary material, note S2; table S4).

Subsequently, we performed the same analysis between *D. plexippus* and *D. melanogaster*. Despite the approximately 580 my of total divergence time between the two species, and the assumed improbability of microsynteny conservation [21], we found 218 cases of microsynteny conservation involving 489 (9.6%) 1-to-1 orthologues. Although the relative presence of these microsynteny blocks was positively correlated with chromosome length in *D. plexippus* ($r^2 = 0.34$, $p = 8.0 \times 10^{-4}$), their presence was not uniform (electronic supplementary material, note S3). Instances involving two 1-to-1 orthologues represented the most common size among microsynteny blocks (185, or 84.9% of the total; electronic supplementary material, table S5). The case involving more 1-to-1 orthologues corresponds to the *Osiris* gene family, relevant for immunity and development [48], in good agreement with previous observations in *B. mori* and other insect lineages [25]. In fact, the Gene Ontology term *Development* was overrepresented through several significantly enriched terms in this subset of orthologues (electronic supplementary material, table S6). Notably, we identified the presence of 16 conserved ancient physical associations corresponding to gene pairs known to be highly refractory to separation across metazoans [3] (electronic supplementary material, table S5).

## (c) Variable dismantling of the ancestral clustering of developmental genes

Clusters of developmental genes often feature unusually high levels of conservation compared to non-development related genes [3,49–51]. This clustering primarily results from

tandem gene duplications [52] subsequently maintained to some degree by regulatory-based constraints. We examined the gene organization of 9 homeobox (Hox, NK, SuperHox, Irx, PRD-LIM and its subcomponent HRO, SINE/Six, and the Vsx and Uncx families) and 5 non-homeobox clusters (Wnt, Fox, Inexin, Runt, E(spl)/Brd) in *D. plexippus* (figure 3), and when necessary in the silkmoth *B. mori* to inform on their differential dynamics of change. All these developmental-related clusters, with the exception of Pharyngeal, Innexin, Runt and E(spl)/Brd, were present in the last common ancestor of deuterostomes and protostomes, while these latter formed more recently [46,50,53,54].

Using 1-to-1 orthologues and TBLASTN sequence similarity searches, we corroborate previous observations relative to clustering patterns for the Hox and Wnt genes in *D. plexippus*, while providing novel information for the rest (electronic supplementary material, note S4). For example, the nine genes forming part of the core of the NK cluster in the bilaterian ancestor (NK5, NK1, Msx, NK4, NK3, Lbx, Tlx, NK7 and Nk6) show many commonalities in their partial clustering in *D. plexippus* and *D. melanogaster*. The orthologues to the *D. melanogaster* genes *HGTX* (NK6) and *NK7.1* (NK7) are approximately 59 kb apart on chromosome *20* of *D. plexippus* whereas those of genes *Dr*, *tin*, *bap* and *lbl/lbe* (Msx, NK4, NK3, Lbx) are found within an approximately 206 kb long interval on chromosome *22*. These two subclusters are also in different chromosomal elements in *D. melanogaster*; the D (*3L*) and E (*3R*), respectively [55]. As in *Apis mellifera* and *Tribolium castaneum* [56], there is one single representative of Lbx in *D. plexippus*. Unexpectedly, *lbe* (Lbx) and *C15* (Tlx), which are contiguous in multiple deuterostomes and protostomes [55], are not in *D. plexippus*, mimicking the conspicuous situation in the distantly related phyla of the tardigrades [57]. This pattern of relative localization among NK genes is also found in *B. mori*. Furthermore, the gene *Dr* is locally duplicated as in *T. castaneum* [58], the two copies being in inverted orientation. Remarkably, in *D. plexippus*, the orthologue to *NK7.1* resides approximately 21 kb away from that to *HHEX*, a member of

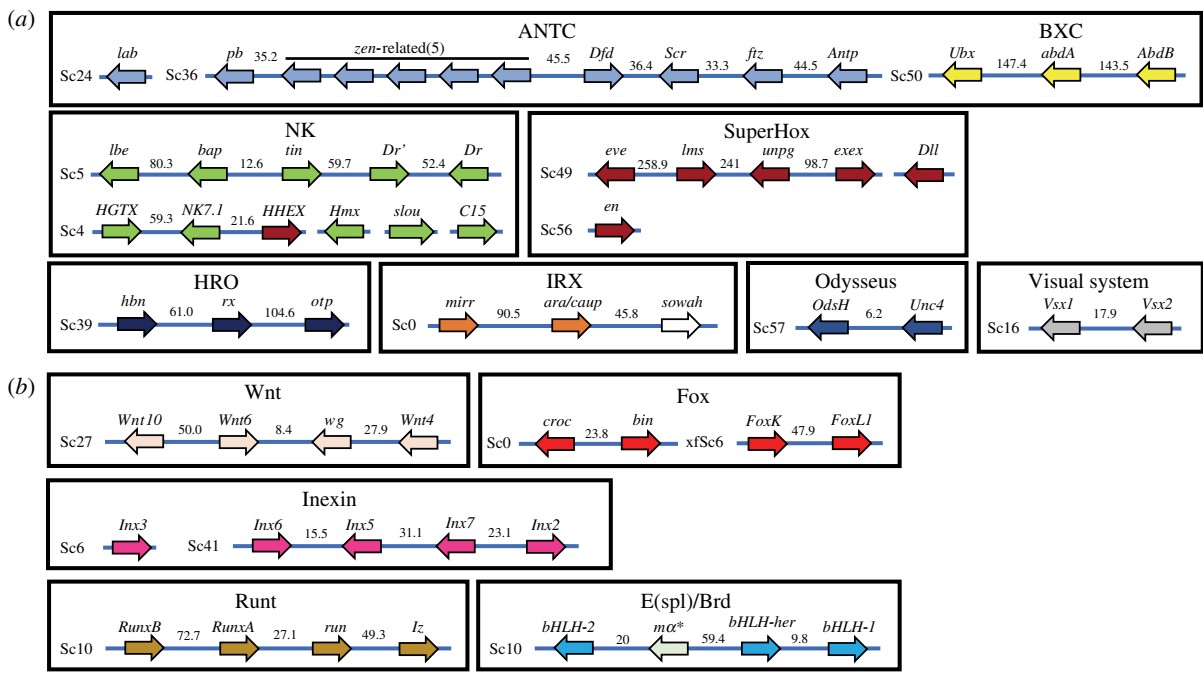

**Figure 3.** Variable dismantling of 12 ancestral developmental gene clusters in the *D. plexipus*. (*a*) Homeobox- and (*b*) non-homeobox-related gene clusters. Gene names as in *D. melanogaster*. In *D. plexippus*, clustering is considered to exist if the distance between relevant genes is lower than 250 kb (i.e. less than 0.1% of the size of the genome assembly; [26]) and regardless of the presence of functionally unrelated genes. The identifier of relevant scaffolds appears abbreviated; for a full identifier see electronic supplementary material, table S7. The only gene not strictly related phylogenetically to the classes of developmental genes considered is *sowah*. It is included here due to its remarkable pattern of conservation in its physical localization relative to other *Irx* genes across Bilateria. The gene classes Pharyngeal, SINE/Six and PRD_LIM do not show evidence of conserved clustering and therefore are not included. *Denotes the orthologue to the Brd-derived gene present in the putative ancestral cluster [53]. (Online version in colour.)

**Table 1.** Overall degree of conservation in gene organization for 14 developmental clusters in *D. plexippus* and *D. melanogaster*. LCA, last common ancestor.

| gene type | cluster | LCA clustering evidence | degree of conservation[a] | |
|---|---|---|---|---|
| | | | *D. plexippus* | *D. melanogaster* |
| homeobox related | Hox | deuterostomes–protostomes | partial (7–8/9)[b] | partial (8/9) |
| | SuperHox | deuterostomes–protostomes | partial (3/6) | absent (0/6) |
| | NK | deuterostomes–protostomes | partial (5/9) | partial (5/9) |
| | Irx | deuterostomes–protostomes | full (2/2) | full (3/3) |
| | Pharyngeal | deuterostomes–protostomes[c] | absent (0/1) | absent (0/1) |
| | SINE | deuterostomes–protostomes | absent (0/2) | absent (0/2) |
| | PRD LIM + subcomponent HRO | deuterostomes–protostomes | absent (0/2) + full (2/2) | absent (0/2) + full (2/2) |
| | Visual system | deuterostomes–protostomes | full (1/1) | full (1/1) |
| | Odysseus | deuterostomes–protostomes | full (1/1) | full (1/1) |
| non-homeobox related | Wnt | deuterostomes–protostomes | full (3/3) | full (3/3) |
| | Fox | deuterostomes–protostomes | partial (2/3) | absent (0/3) |
| | Innexin | invertebrates | partial (3/4) | partial (1/4) |
| | Runt | insect | full (3/3) | partial (2/3) |
| | E(spl)/Brd | crustacean–insect | full (3/3) | partial (13/15) |

[a]In parenthesis, the number of conserved contiguities among genes of the same cluster in relation to the total number of contiguities based on the number of genes part of such cluster. See electronic supplementary material, table S7 for more details.

[b]Variable number of conserved contiguities depending on the information considered. In the assembly DpMex_v1 [26], the genes *Antp* and *Ubx* are at the end of the scaffolds Sc000036 and Sc000005, respectively, which precludes to confirm whether or not they are contiguous. A previous analysis in *D. plexippus* indicated that *Antp* was contiguous to *Ubx* [5].

[c]The pharyngeal gene cluster has been reported in a variety of deuterostomes while only a few of its constituent genes were found to cluster in one non-deuterostome examined [46]. Although one of the genes of this cluster, Pax1/9, currently encodes a peptide that lacks a homeobox domain, its ancestral version is thought to have included this type of domain [49].

the SuperHox ancestral cluster, without evidence of any intervening gene (figure 3). Positional information of the orthologue to *HHEX* relative to that of NK genes in the deuterostomes *Branchioma floridae*, *Saccoglossus kowalevskii* and *Ptychodera flava* led to propose an ancient association of the SuperHox and NK clusters in the ancestor to the Bilateria [46,49]. Our results unambiguously support this hypothesis by confirming the linkage between these two subclasses of homeobox gene families in a protostome.

Overall, only four (the smallest, with two genes, and therefore more unlikely to be disrupted) out of 14 ancient clusters (Irx, Visual system, Odysseus, and Wnt) appear undisrupted by large-scale structural rearrangements in *D. plexippus* and *D. melanogaster* (table 1; electronic supplementary material, table S7; figure 3). The 10 remaining clusters present some level of decay either in one or both lineages, which can be accounted for by several non-mutually exclusive factors. The first is the absence of global functional constraints operating on cluster architecture (i.e. microsynteny conservation would be just reflecting phylogenetic inertia). The second factor is that regulatory-based constraints, if they ever existed, could have been lifted in concert with the evolution of lineage-specific developmental properties [1]. The third factor is the limited fitness cost that structural remodelling might have in some insect lineages, which agrees well with the high malleability of their genomes. These limited detrimental effects could stem from an achiasmatic meiosis in the heterogametic sex in both *D. plexippus* and *D. melanogaster*, the presence of holocentric chromosomes in *D. plexippus,* and compensatory mechanisms during *D. melanogaster* meiosis [18,35,59].

## 4. Discussion

Chromosomal rearrangements can play pivotal roles in environmental adaptation, phenotypic diversification and reproductive isolation [60,61]. Nevertheless, the extent to which the chromosomal gene content and organization has been remodelled during species divergence, particularly among distantly related and structurally dynamic genomes, remains elusive due to limited contiguity of genome assemblies, suboptimal or entirely absent gene annotations, as well as non-reliable orthologues relationships. By examining two species from the Lepidoptera and Diptera with reference-quality genome assemblies, enhanced gene sets and reliable orthologues relationships [26,62], we found evidence that a presumed quasi-randomization of the gene organization [4,21] at a fine scale between the species compared is far from complete. Additionally, at the level of whole-chromosome organization, we have extended and refined previous inferences based on a more limited number of orthologues landmarks [21], showing the marked signatures of macrosynteny conservation between the chromosomes of *D. plexippus* and *D. melanogaster*. These signatures strongly support a virtual absence of large-scale exchanges of gene content between ancestral chromosomes during and after the formation of the protokaryotypes of both Lepidoptera and Diptera.

Collectively, our results provide a glimpse of how some portions of that ancestral genome to the Lepidoptera and Diptera were organized. As a result, attempts to reconstruct ancestral states have the potential to help uncover the chromosomal gene organization and content not only in the ancestor to the Insecta but also in that of the Bilateria as has been the case here for the SuperHox and NK gene clusters. The forthcoming availability of quality-reference genome assemblies, enhanced gene sets and reliable orthologues relationships in other insect orders, as well as in additional protostomes and deuterostomes, will facilitate this task while providing further insights into how and to what extent chromosome gene organization and content have changed at different phylogenetic distances.

Rates of gene order evolution in the Lepidoptera and Diptera orders have been occasionally compared, inferring that the rates in the former are comparable or even higher than those in the *Drosophila* genus, arguably because of the holocentric nature of the Lepidoptera chromosomes [5,18,63]. By using genome assemblies with relatively low fragmentation, we find that, if anything, the remodelling rates of gene organization are in fact lower in Lepidoptera than in the *Drosophila* genus. In good agreement, our systematic analysis of clusters of developmental genes allowed us to test this trend at a finer scale, particularly as these clusters are more likely under functional constraints due to complex gene regulation. Based on the 10 clusters that showed some degree of decay, inter-lineage differences are evident for five of them (SuperHox, Fox, Innexin, Runt and E(spl)/Brd), with *D. plexippus* systematically showing an overall increased level of clustering conservation relative to *D. melanogaster*. This observation could be explained not only by a slightly higher rate of chromosome repatterning in the Diptera but also by an increased local refractoriness to the occurrence of breakpoints in genomic regions of *D. plexippus* that are likely to be exposed to different regulatory inputs. This increased refractoriness would result in a less uniform distribution of breakpoints of structural rearrangements along the Lepidoptera chromosomes. Future guided genome remodelling in insect species beyond *D. melanogaster* [64] will be instrumental to disentangle the relative contribution of these alternative explanations.

**Data accessibility.** Lists of 1-to-1 orthologues, including their molecular coordinates and chromosomal locations, were retrieved from the Dryad Digital Repository: https://doi.org/10.7280/D1WM43 [65].

The data are provided in electronic supplementary material [66].

**Authors' contributions.** J.M.R.: conceptualization, data curation, formal analysis, funding acquisition, investigation, methodology, project administration, resources, software, supervision, validation, visualization, writing-original draft, writing-review and editing; P.M.G.: data curation; R.N.S.: data curation, formal analysis; S.J.B.: data curation, formal analysis; C.A.-G.: conceptualization, funding acquisition; T.M.: conceptualization, validation, writing-review and editing.

All authors gave final approval for publication and agreed to be held accountable for the work performed therein.

**Competing interests.** The authors declare no conflict of interest.

**Funding.** This work was supported by a UCI Mexus grant to C.A.-G. and J.M.R.

**Acknowledgements.** We thank Igor Sharakhov and Stephen Schaeffer for feedback on early versions of the manuscript.

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
