## [Peer Review File · Proceedings of the Royal Society B: Biological Sciences]

Review History

RSPB-2021-1561.R0 (Original submission)

Review form: Reviewer 1

Recommendation

Accept with minor revision (please list in comments)

Scientific importance: Is the manuscript an original and important contribution to its field?

Excellent

General interest: Is the paper of sufficient general interest?

Good

Quality of the paper: Is the overall quality of the paper suitable?

Good

Is the length of the paper justified?

Yes

Should the paper be seen by a specialist statistical reviewer?

Yes

Do you have any concerns about statistical analyses in this paper? If so, please specify them explicitly in your report.

No

It is a condition of publication that authors make their supporting data, code and materials available - either as supplementary material or hosted in an external repository. Please rate, if applicable, the supporting data on the following criteria.

Is it accessible?

Yes

Is it clear?

Yes

Is it adequate?

Yes

Do you have any ethical concerns with this paper?

No

Comments to the Author

The authors compare the genome-wide gene organisation across three insects: a butterfly, moth and fruit fly. This work capitalises on the recent improvement in the quality of the genome assembly of this butterfly and represents exactly the sort of research that is being facilitated with the recent improvements in genome sequencing and assembly, and that is urgently needed for us to obtain a proper understanding of the dynamics of genome evolution and organisation.

The relatively high rates of chromosome reorganisation in these two insect lineages (e.g. chromosome fusion and fission in both Lepidoptera and Diptera) add to the interest of these specific comparisons, and reveal the rather surprising finding, given these high rearrangement rates, that extensive levels of synteny have been conserved.

The research includes an interesting, innovative use of the tau index, repurposing it for gene linkage rather than expression. Also, the supplementary information is a treasure-trove of information, providing excellent context for the main figures. I look forward to this approach being applied more widely in future work, to establish whether these levels of synteny conservation, although much higher than expected relative to randomised genomes, actually are relatively low compared to other genomes or not.

Overall, this manuscript is timely and innovative, well-written and of wide potential interest.

My comments are all minor in nature.

1) In the Methods, I was unclear why the authors removed the 95 genes that are implicated in interchromosomal translocations in Lepidoptera, but then (top line of page 9) say that comparison between the two lepidopterans shows no interchromosomal exchange?

2) In Table 1, why are the Irx clusters listed as 'Partial' when they seem to contain all of the relevant genes in these species?

3) The coverage of the Pharyngeal cluster is a little confusing. Table 1 lists this as a protostome cluster. However, I presumed that the authors were talking about the pharyngeal cluster described in Simakov et al (2015) [ref 43], which is confirmed by the supplementary text that explains things more clearly and correctly. Simakov describe this cluster mainly for

deuterostomes, with only elements of smaller parts of the cluster seeming to be present in one of the non-deuterostomes that they checked. This sort of confusion might be resolved to some extent by including relevant citations in the paragraph on pages 9-10 adjacent to the various gene clusters that are mentioned. This cluster also contains a mix of homeobox-containing and non-homeobox genes, so which category should it be listed under in the text on page 9 and in Table 1?

4) The sentence spanning pages 10-11 is quite hard to read due to the complex punctuation (with colons, semi-colons and commas). This needs rewriting and breaking up into distinct sentences.

5) Figure 3: why is the BX-C component of the Hox cluster shown in the order AbdB-abda-Ubx, when it is known that the ancestral order would have had these three genes in the reverse order relative to the rest of the cluster? Since they are on a distinct scaffold relative to the ANT-C components then there seems no reason to suggest a derived inverted organisation.

Review form: Reviewer 2

Recommendation

Reject – article is not of sufficient interest (we will consider a transfer to another journal)

Scientific importance: Is the manuscript an original and important contribution to its field?

Acceptable

General interest: Is the paper of sufficient general interest?

Marginal

Quality of the paper: Is the overall quality of the paper suitable?

Acceptable

Is the length of the paper justified?

Yes

Should the paper be seen by a specialist statistical reviewer?

No

Do you have any concerns about statistical analyses in this paper? If so, please specify them explicitly in your report.

No

It is a condition of publication that authors make their supporting data, code and materials available - either as supplementary material or hosted in an external repository. Please rate, if applicable, the supporting data on the following criteria.

Is it accessible?

Yes

Is it clear?

Yes

Is it adequate?

Yes

Do you have any ethical concerns with this paper?

No

Comments to the Author

The authors show that macrosynteny and some microsynteny is conserved between Diptera and Lepidoptera, and investigate some interesting cases of developmental gene clusters. These data are a nice contribution, well explained, and technically appropriate as far as I can see.

Unfortunately, however, I am not persuaded that the contribution raises to the level of general interest expected by Proceedings B. That synteny has not been fully scrambled is basically a statement that rate-of-change X time is less than some value (saturation, basically). Given that that has been shown over much longer evolutionary times in different lineages (non-bilaterian animals X bilaterian animals, for example), I do not find this finding surprising in important ways (that previous analyses failed to find the result does not make overturning the result particularly important unless the result itself is particularly important). Findings that gene clusters are partially conserved is also a worthy contribution but it is not clear what this finding qualitatively contributes to our understanding to the underlying biological questions.

As such, I would strongly support the manuscript for publication in a different journal, but am not convinced that it is sufficiently groundbreaking for Proceedings B. I am sorry that I cannot to be more positive on this occasion.

Two points:

First, the authors might consider whether it would be possible to learn more about the features of gene linkages that are more likely to be maintained -- can they see a signature of differences in rates of rearrangement? (I'm not sure this would be interesting enough to justify the effort, since much of what one might find could be just distances between genes (larger intergenic regions predict greater rearrangement rates, for example), but perhaps worth considering.

While the paper is generally clearly written, the section beginning "To further quantify the specificity of association between the gene contents" is less so and could benefit from a rewrite (τ is not clear, the paragraph appears to include unrelated results, etc.)

Decision letter (RSPB-2021-1561.R0)

07-Sep-2021

Dear Professor Ranz:

I am writing to inform you that we have now obtained responses from referees on manuscript RSPB-2021-1561 entitled "Multiscale analysis of the randomization limits of the chromosomal gene organization between Lepidoptera and Diptera." which you submitted to Proceedings B.

Unfortunately, on the advice of the Associate Editor and the referees, your manuscript has been rejected following full peer review. Competition for space in Proceedings B is currently extremely severe, as many more manuscripts are submitted to us than we have space to print. We are therefore only able to publish those that are exceptional, convincing and present significant advances of broad interest, and must reject many good manuscripts.

Please find below the comments received from referees concerning your manuscript, not including confidential reports to the Editor. I hope you may find these useful should you wish to submit your manuscript elsewhere.

We are sorry that your manuscript has had an unfavourable outcome, but would like to thank you for offering your work to Proceedings B.

Sincerely,
Dr Locke Rowe
mailto: proceedingsb@royalsociety.org

Associate Editor
Board Member: 1
Comments to Author:

Both reviewers were impressed by the clever and novel approaches used here to characterize in detail the conservation of synteny in and between Lepidoptera and Diptera. I agree that this is a strong and well-written manuscript on a timely topic. However, the reviewers also point out that the current work does not investigate whether conservation of synteny is unusually high in this clade, or get at what forces shape the rates of genome rearrangements in general, making it a better fit for a more specialized journal.

Reviewer(s)' Comments to Author:
Referee: 1

Comments to the Author(s)

The authors compare the genome-wide gene organisation across three insects: a butterfly, moth and fruit fly. This work capitalises on the recent improvement in the quality of the genome assembly of this butterfly and represents exactly the sort of research that is being facilitated with the recent improvements in genome sequencing and assembly, and that is urgently needed for us to obtain a proper understanding of the dynamics of genome evolution and organisation.

The relatively high rates of chromosome reorganisation in these two insect lineages (e.g. chromosome fusion and fission in both Lepidoptera and Diptera) add to the interest of these specific comparisons, and reveal the rather surprising finding, given these high rearrangement rates, that extensive levels of synteny have been conserved.

The research includes an interesting, innovative use of the tau index, repurposing it for gene linkage rather than expression. Also, the supplementary information is a treasure-trove of information, providing excellent context for the main figures. I look forward to this approach being applied more widely in future work, to establish whether these levels of synteny conservation, although much higher than expected relative to randomised genomes, actually are relatively low compared to other genomes or not.

Overall, this manuscript is timely and innovative, well-written and of wide potential interest.

My comments are all minor in nature.

- 1) In the Methods, I was unclear why the authors removed the 95 genes that are implicated in interchromosomal translocations in Lepidoptera, but then (top line of page 9) say that comparison between the two lepidopterans shows no interchromosomal exchange?
- 2) In Table 1, why are the Irx clusters listed as 'Partial' when they seem to contain all of the relevant genes in these species?
- 3) The coverage of the Pharyngeal cluster is a little confusing. Table 1 lists this as a protostome cluster. However, I presumed that the authors were talking about the pharyngeal cluster described in Simakov et al (2015) [ref 43], which is confirmed by the supplementary text that explains things more clearly and correctly. Simakov describe this cluster mainly for deuterostomes, with only elements of smaller parts of the cluster seeming to be present in one of the non-deuterostomes that they checked. This sort of confusion might be resolved to some extent by including relevant citations in the paragraph on pages 9-10 adjacent to the various gene

clusters that are mentioned. This cluster also contains a mix of homeobox-containing and non-homeobox genes, so which category should it be listed under in the text on page 9 and in Table 1?

4) The sentence spanning pages 10-11 is quite hard to read due to the complex punctuation (with colons, semi-colons and commas). This needs rewriting and breaking up into distinct sentences.

5) Figure 3: why is the BX-C component of the Hox cluster shown in the order *AbdB-abda-Ubx*, when it is known that the ancestral order would have had these three genes in the reverse order relative to the rest of the cluster? Since they are on a distinct scaffold relative to the ANT-C components then there seems no reason to suggest a derived inverted organisation.

Referee: 2

Comments to the Author(s)

The authors show that macrosynteny and some microsynteny is conserved between Diptera and Lepidoptera, and investigate some interesting cases of developmental gene clusters. These data are a nice contribution, well explained, and technically appropriate as far as I can see.

Unfortunately, however, I am not persuaded that the contribution raises to the level of general interest expected by Proceedings B. That synteny has not been fully scrambled is basically a statement that rate-of-change \times time is less than some value (saturation, basically). Given that that has been shown over much longer evolutionary times in different lineages (non-bilaterian animals \times bilaterian animals, for example), I do not find this finding surprising in important ways (that previous analyses failed to find the result does not make overturning the result particularly important unless the result itself is particularly important). Findings that gene clusters are partially conserved is also a worthy contribution but it is not clear what this finding qualitatively contributes to our understanding to the underlying biological questions.

As such, I would strongly support the manuscript for publication in a different journal, but am not convinced that it is sufficiently groundbreaking for Proceedings B. I am sorry that I cannot to be more positive on this occasion.

Two points:

First, the authors might consider whether it would be possible to learn more about the features of gene linkages that are more likely to be maintained -- can they see a signature of differences in rates of rearrangement? (I'm not sure this would be interesting enough to justify the effort, since much of what one might find could be just distances between genes (larger intergenic regions predict greater rearrangement rates, for example), but perhaps worth considering.

While the paper is generally clearly written, the section beginning "To further quantify the specificity of association between the gene contents" is less so and could benefit from a rewrite (τ is not clear, the paragraph appears to include unrelated results, etc.)

Author's Response to Decision Letter for (RSPB-2021-1561.R0)

See Appendix A.

RSPB-2021-2183.R0

Review form: Reviewer 2

Recommendation

Reject – article is not of sufficient interest (we will consider a transfer to another journal)

Scientific importance: Is the manuscript an original and important contribution to its field?

Marginal

General interest: Is the paper of sufficient general interest?

Marginal

Quality of the paper: Is the overall quality of the paper suitable?

Good

Is the length of the paper justified?

Yes

Should the paper be seen by a specialist statistical reviewer?

Yes

Do you have any concerns about statistical analyses in this paper? If so, please specify them explicitly in your report.

No

It is a condition of publication that authors make their supporting data, code and materials available - either as supplementary material or hosted in an external repository. Please rate, if applicable, the supporting data on the following criteria.

Is it accessible?

No

Is it clear?

No

Is it adequate?

No

Do you have any ethical concerns with this paper?

No

Comments to the Author

Unfortunately, neither the new draft nor the response to my comments changes my evaluation of the manuscript substantially. I do not understand why the authors don't think that conservation between more deeply-diverged animals diminishes the significance of their finding of conservation between more recently-diverged animals. I do not see that any important general statements about biology hinge on the rate of rearrangements specifically in insects. Therefore, I unfortunately still do not support publication in Proc B.

Review form: Reviewer 3

Recommendation

Accept as is

Scientific importance: Is the manuscript an original and important contribution to its field?

Good

General interest: Is the paper of sufficient general interest?

Good

Quality of the paper: Is the overall quality of the paper suitable?

Excellent

Is the length of the paper justified?

Yes

Should the paper be seen by a specialist statistical reviewer?

No

Do you have any concerns about statistical analyses in this paper? If so, please specify them explicitly in your report.

No

It is a condition of publication that authors make their supporting data, code and materials available - either as supplementary material or hosted in an external repository. Please rate, if applicable, the supporting data on the following criteria.

Is it accessible?

Yes

Is it clear?

Yes

Is it adequate?

Yes

Do you have any ethical concerns with this paper?

No

Comments to the Author

This is an interesting paper that tries to begin to unravel the nature of the fast genome rearrangement in the insect genomes. The authors have responded to the previous reviewers well and I would thus recommend this paper for publication.

Decision letter (RSPB-2021-2183.R0)

07-Dec-2021

Dear Professor Ranz

I am pleased to inform you that your manuscript RSPB-2021-2183 entitled "Multiscale analysis of the randomization limits of the chromosomal gene organization between Lepidoptera and Diptera." has been accepted for publication in Proceedings B.

The referee(s) have recommended publication, but also suggest some minor revisions to your manuscript. Therefore, I invite you to respond to the referee(s)' comments and revise your manuscript. Because the schedule for publication is very tight, it is a condition of publication that you submit the revised version of your manuscript within 7 days. If you do not think you will be able to meet this date please let us know.

In order to ensure effective and robust dissemination and appropriate credit to authors the dataset(s) used should be fully cited. To ensure archived data are available to readers, authors

should include a 'data accessibility' section immediately after the acknowledgements section. This should list the database and accession number for all data from the article that has been made publicly available, for instance:

If you wish to submit your data to Dryad (<http://datadryad.org/>) and have not already done so you can submit your data via this link [http://datadryad.org/submit?journalID=RSPB&manu=\(Document not available\)](http://datadryad.org/submit?journalID=RSPB&manu=(Document%20not%20available)) which will take you to your unique entry in the Dryad repository. If you have already submitted your data to dryad you can make any necessary revisions to your dataset by following the above link. Please see <https://royalsociety.org/journals/ethics-policies/data-sharing-mining/> for more details.

Sincerely,
Dr Locke Rowe
mailto:proceedingsb@royalsociety.org

Associate Editor
Board Member
Comments to Author:

The revised manuscript has been seen by reviewer 2 and by a third expert. None of the reviewers have concerns about the analysis, although reviewer 2 still feels that a stronger case could be made for the general interest of the paper. You may wish to address this before resubmission (which I would not expect to be sent out for review again).

Reviewer(s)' Comments to Author:

Referee: 2

Comments to the Author(s).

Unfortunately, neither the new draft nor the response to my comments changes my evaluation of the manuscript substantially. I do not understand why the authors don't think that conservation between more deeply-diverged animals diminishes the significance of their finding of conservation between more recently-diverged animals. I do not see that any important general statements about biology hinge on the rate of rearrangements specifically in insects. Therefore, I unfortunately still do not support publication in Proc B.

Referee: 3

Comments to the Author(s).

This is an interesting paper that tries to begin to unravel the nature of the fast genome rearrangement in the insect genomes. The authors have responded to the previous reviewers well and I would thus recommend this paper for publication.

Author's Response to Decision Letter for (RSPB-2021-2183.R0)

See Appendix B.

Decision letter (RSPB-2021-2183.R1)

13-Dec-2021

Dear Professor Ranz

I am pleased to inform you that your manuscript entitled "Multiscale analysis of the randomization limits of the chromosomal gene organization between Lepidoptera and Diptera." has been accepted for publication in Proceedings B.

If you are likely to be away from e-mail contact please let us know. Due to rapid publication and an extremely tight schedule, if comments are not received, we may publish the paper as it stands. If you have any queries regarding the production of your final article or the publication date please contact procb_proofs@royalsociety.org

Your article has been estimated as being 9 pages long. Our Production Office will be able to confirm the exact length at proof stage.

Data Accessibility section

Open Access

Paper charges

Sincerely,
Editor, Proceedings B
<mailto:proceedingsb@royalsociety.org>

Appendix A

Reviewer(s)' Comments to Author:

Referee: 1

Comments to the Author(s)

The authors compare the genome-wide gene organisation across three insects: a butterfly, moth and fruit fly. This work capitalises on the recent improvement in the quality of the genome assembly of this butterfly and represents exactly the sort of research that is being facilitated with the recent improvements in genome sequencing and assembly, and that is urgently needed for us to obtain a proper understanding of the dynamics of genome evolution and organisation.

The relatively high rates of chromosome reorganisation in these two insect lineages (e.g. chromosome fusion and fission in both Lepidoptera and Diptera) add to the interest of these specific comparisons, and reveal the rather surprising finding, given these high rearrangement rates, that extensive levels of synteny have been conserved.

The research includes an interesting, innovative use of the tau index, repurposing it for gene linkage rather than expression. Also, the supplementary information is a treasure-trove of information, providing excellent context for the main figures. I look forward to this approach being applied more widely in future work, to establish whether these levels of synteny conservation, although much higher than expected relative to randomized genomes, actually are relatively low compared to other genomes or not.

Overall, this manuscript is timely and innovative, well-written and of wide potential interest.

My comments are all minor in nature.

1) In the Methods, I was unclear why the authors removed the 95 genes that are implicated in interchromosomal translocations in Lepidoptera, but then (top line of page 9) say that comparison between the two lepidopterans shows no interchromosomal exchange?

The reviewer is correct in that, in the way the information is provided, there seems to be a discrepancy. In the sentence of the Results section (now lines 218-219), we meant that there was no large-scale interchromosomal exchange. Interchromosomal gene transposition are not considered large-scale interchromosomal exchanges.

We have rewritten the sentences in the Methods and Results sections; the modified text appears highlighted in yellow. In the first case, it reads:

“Ninety-five orthologs potentially involved in interchromosomal gene transpositions between *D. plexippus* and *B. mori* (Ranz et al. 2021) were not considered in downstream analyses as they are not part of large-scale chromosomal rearrangements.”

In the second, it reads:

“These microsynteny blocks would have been reshuffled within the contemporary versions of ancestral chromosomes by 308-337 inversions (Supplementary Note 1), disrupting the overall collinearity along the chromosomes of these species to different extents, although in no case involving large-scale exchanges of gene content between different chromosomes, including those that are composites of two ancestral chromosomes due to fusion events (Fig. 2).”

2) In Table 1, why are the lrx clusters listed as ‘Partial’ when they seem to contain all of the relevant genes in these species?

The reviewer is correct, and the inaccuracy has been fixed in Table 1 and in the text (lines 266-267). The number of clusters whose contiguities are conserved both in *Danaus* and *Drosophila* is therefore 4 and not 3 as we indicated in the main text of our initial submission.

3) The coverage of the Pharyngeal cluster is a little confusing. Table 1 lists this as a protostome cluster. However, I presumed that the authors were talking about the pharyngeal cluster described in Simakov et

al (2015) [ref 43], which is confirmed by the supplementary text that explains things more clearly and correctly. Simakov describe this cluster mainly for deuterostomes, with only elements of smaller parts of the cluster seeming to be present in one of the non-deuterostomes that they checked. This sort of confusion might be resolved to some extent by including relevant citations in the paragraph on pages 9-10 adjacent to the various gene clusters that are mentioned. This cluster also contains a mix of homeobox-containing and non-homeobox genes, so which category should it be listed under in the text on page 9 and in Table 1?

The reviewer is correct in his/her interpretation about what we meant. We have followed the advice citing Simakov et al. (2015) and other relevant references to resolve the issue (now line 246). In addition, in Table 1, we have added a footnote that explains the peculiarities of this cluster in deuterostomes and protostomes. Further, the reviewer makes a very important observation about the Pharyngeal cluster as it includes both homeobox and non-homeobox encoding genes. We have opted for leaving this cluster in the Homeobox related category and added further details to the indicated footnote, explaining that Pax1/9 currently encodes a peptide that lacks a homeobox domain although its ancestral version is thought to have included this type of domain. Let us know if you have a better suggestion.

4) The sentence spanning pages 10-11 is quite hard to read due to the complex punctuation (with colons, semi-colons and commas). This needs rewriting and breaking up into distinct sentences. We have followed reviewer's indication and split the long sentence. Now it reads:

“Overall, only four –the smallest with two genes and therefore more unlikely to be disrupted– out of 14 ancient clusters (*lrx*, Visual system, Odysseus, and Wnt) appear undisrupted by large-scale structural rearrangements in *D. plexippus* and *D. melanogaster* (Table 1 and Supplementary Table 7; Fig. 3). The 10 remaining clusters present some level of decay either in one or both lineages, which can be accounted by several non-mutually exclusive factors. The first is the absence of global functional constraints operating on cluster architecture, *i.e.* microsynteny conservation would be just reflecting phylogenetic inertia. The second factor is that regulatory-based constraints, if ever existed, could have been lifted in concert with the evolution of lineage-specific developmental properties (Duboule 2007). The third factor is the limited fitness cost that structural remodeling might have in some insect lineages, which agrees well with the highly malleability of their genomes. These limited detrimental effects could stem from an achiasmatic meiosis in the heterogametic sex in both *D. plexippus* and *D. melanogaster*, the presence of holocentric chromosomes in *D. plexippus*, and compensatory mechanisms during *D. melanogaster* meiosis (Carlson 1946; d’Alençon et al. 2010; Mongue et al. 2017).”

5) Figure 3: why is the BX-C component of the Hox cluster shown in the order AbdB-abda-Ubx, when it is known that the ancestral order would have had these three genes in the reverse order relative to the rest of the cluster? Since they are on a distinct scaffold relative to the ANT-C components then there seems no reason to suggest a derived inverted organisation.

The reviewer is correct in his/her observation and we have redrawn the figure accordingly.

Referee: 2

Comments to the Author(s)

The authors show that macrosynteny and some microsynteny is conserved between Diptera and Lepidoptera, and investigate some interesting cases of developmental gene clusters. These data are a nice contribution, well explained, and technically appropriate as far as I can see.

Unfortunately, however, I am not persuaded that the contribution raises to the level of general interest expected by Proceedings B. That synteny has not been fully scrambled is basically a statement that rate-of-change X time is less than some value (saturation, basically). Given that that has been shown over much longer evolutionary times in different lineages (non-bilaterian animals X bilaterian animals, for example), I do not find this finding surprising in important ways (that previous analyses failed to find the result does not make overturning the result particularly important unless the result itself is particularly important). Findings that gene clusters are partially conserved is also a worthy contribution but it is not clear what this finding qualitatively contributes to our understanding to the underlying biological questions.

As such, I would strongly support the manuscript for publication in a different journal, but am not convinced that it is sufficiently groundbreaking for Proceedings B. I am sorry that I cannot be more positive on this occasion.

We respectfully dissent from the reviewer. He/she should consider the following:

1. Finding evidence of synteny conservation between *Danaus* and *Drosophila* is not comparable in any way with previous observations involving non-bilaterian and bilaterian organisms. Nematodes and insects exhibit the highest rates of chromosome evolution ever documented (Mitreva et al. 2005; von Grotthuss et al. 2010; Heliconius Genome et al. 2012; Neafsey et al. 2015). This, coupled with a completely different dynamics in fusions and fissions (the first prevailing in the lineage to Diptera and the second in the that to Lepidoptera), and possible occurrence of large-scale interchromosomal exchange during a fraction of their divergence time represents a formidable test to synteny conservation.

2. What we provide is an integral analysis about the limits of chromosome rearrangements between 2 species that belong to two different insect orders known for possessing extremely high rates of gene order evolution (only matched by the ones in nematodes) and currently with unparalleled genomic resources. Therefore, this work is pioneering in paving the way for others once those high-quality resources become available.

We would like to note as well that although in recent years the number of highly contiguous genome assemblies has remarkably surged (in and out the Insecta), more often than not, these assemblies lack of any gene annotation (Hotaling et al. 2021; Genome Biology and Evolution; doi:10.1093/gbe/evab138), and if they have, their annotation is far from the comprehensiveness and reliability of the one we have generated for *D. plexippus* (Ranz et al. 2021). Put simply: at this moment, replicating an equally reliable and precise comparative analysis to the one we have performed is non-trivial, particularly among highly malleable genomes. We believe this has its own merit.

3. We also find evidence of microsynteny conservation, which is not mentioned by the reviewer. This is another exceptional finding and clearly in conflict with what other eminent colleagues have reported previously. For example, Pease and Hahn in Molecular Biology and Evolution (<https://doi.org/10.1093/molbev/mss010>) stated:

“Many previous studies of chromosomal homology have taken the spatial relationship of genes into account (i.e., the synteny of contiguous gene blocks). Because very high rates of rearrangement within single insect chromosomes can occur even within a genus (e.g., *Drosophila*; Bhutkar et al. 2008), resolving these movements among species in distantly diverged orders is likely a near impossibility.”

Therefore, finding microsynteny conservation beyond some anecdotal cases at these phylogenetic distance -as others have done in the past- between two highly dynamic genomes and providing, to the best of our knowledge, the first reliable quantification of microsynteny conservation between insect orders in the literature is exceptional.

In sum, we believe that collectively our results, their precision and reliability, do actually make an important contribution to comparative genomics -in and out the Insecta- and therefore fit perfectly well with Proceedings B. Our work is timely, solid, and rigorous, serving as a baseline for others to come. And this while avoiding unnecessary flashiness. We have largely rewritten the Introduction and Discussion to be more explicit about the justification of our work and the broad implications of our findings.

Two points:

First, the authors might consider whether it would be possible to learn more about the features of gene linkages that are more likely to be maintained -- can they see a signature of differences in rates of rearrangement? (I'm not sure this would be interesting enough to justify the effort, since much of what one might find could be just distances between genes (larger intergenic regions predict greater rearrangement rates, for example), but perhaps worth considering.

We again must respectfully dissent from the reviewer. In the case of highly malleable genomes at the structural level (references noted above), the rationale proposed by the reviewer is only suitable between very closely related species. Take as an example the genus *Drosophila*. Given the extremely high rates of chromosome repatterning -mostly via paracentric inversions- and differential TE dynamics and composition between species, there exist a tremendous reshaping of intergenic regions, making extremely complex to establish ancestral states for such regions in terms of length and therefore to perform the analysis suggested.

Further, the reviewer should consider that large intergenic regions outside the heterochromatin are often reflecting at least one of these two scenarios:

1. Regions populated with cis-regulatory modules that impact the expression of nearby genes. These regions, often referred to as "gene deserts" are highly refractory to chromosomal rearrangements. Multiple genes involved in development and signaling pathways reside in gene deserts.
2. Regions not properly annotated. These regions, once they are annotated with for example the addition of RNA-seq data, appear populated with additional genes. In other words, the test that the reviewer proposes can only be done with species possessing unmatched genome resources for which it is possible to be certain about the length of the intergenic regions. For example, this is not possible yet even for most *Drosophila* species.

An example that illustrates both points 1 and 2 above is that involving the genes *C15* and one *ladybird* paralog. These two genes are part of the NK developmental complex. Even in *D. melanogaster*, species with a quite compact genome, these two genes are separated by 55 kb -quite large for the euchromatin. More recent gene annotations have shown the presence of two lncRNA and one protein-coding gene in between the mentioned NK genes. The physical association of these two genes has been confirmed in all *Drosophila* species examined so far as well as in *An. gambiae* (Chan et al. 2015).

In sum, we believe that only with a more precise annotation of genes and regulatory sequences across multiple Diptera and Lepidoptera genomes one can try to implement the test suggested by the reviewer.

While the paper is generally clearly written, the section beginning "To further quantify the specificity of association between the gene contents" is less so and could benefit from a rewrite (tau is not clear, the paragraph appears to include unrelated results, etc.)

We agree with the reviewer that the indicated paragraph can, and must, be improved. We have split the paragraph in two and modified slightly the wording to enhance internal coherence (lines 184-198).

Relative to the tau index, the reviewer is right in that it requires a better explanation as not all the readers may be necessarily well versed on it. Accordingly, we now provide a more precise description of its use in the Methods. Specifically, we say:

"The breadth of gene content association of a given *D. plexippus* chromosome relative to the *D. melanogaster* chromosomes was calculated using the τ index (Yanai et al. 2005), which takes values from 0 to 1, and it is defined as

$$\tau = \frac{\sum_{i=1}^N (1 - x_i)}{N - 1}$$

where N is the number of chromosomal elements in *D. melanogaster* (X, 2L, 2R, 3L, 3R, and 4) and x_i is the number 1-to-1 orthologs located in each *D. melanogaster* chromosome normalized by the maximum number. In this case, τ measures the degree of location specificity of 1-to-1 orthologs between the chromosomes of the two species. The higher the value, the larger the fraction of the orthologs harbored by a given chromosome of *D. plexippus* that reside in each *Drosophila* chromosomal element. To avoid any distorting effect associated with the comparably limited number of orthologs located on the dot-like chromosome 4 of *D. melanogaster* in relation to the rest, we repeated the analysis excluding this chromosome.”

References

- Chan C, Jayasekera S, Kao B, Paramo M, von Grotthuss M, Ranz JM. 2015. Remodelling of a homeobox gene cluster by multiple independent gene reunions in *Drosophila*. *Nat Commun* 6:6509.
- Heliconius Genome C, Dasmahapatra KK, Walters JR, Briscoe AD, Davey JW, Whibley A, Nadeau NJ, Zimin AV, Hughes DST, Ferguson LC, et al. 2012. Butterfly genome reveals promiscuous exchange of mimicry adaptations among species. *Nature* 487:94-98.
- Mitreva M, Blaxter ML, Bird DM, McCarter JP. 2005. Comparative genomics of nematodes. *Trends in Genetics* 21:573-581.
- Neafsey DE, Waterhouse RM, Abai MR, Aganezov SS, Alekseyev MA, Allen JE, Amon J, Arca B, Arensburger P, Artemov G, et al. 2015. Mosquito genomics. Highly evolvable malaria vectors: the genomes of 16 *Anopheles* mosquitoes. *Science* 347:1258522.
- Ranz JM, Gonzalez PM, Clifton BD, Nazario-Yepiz NO, Hernandez-Cervantes PL, Palma-Martinez MJ, Valdivia DI, Jimenez-Kaufman A, Lu MM, Markow TA, et al. 2021. A de novo transcriptional atlas in *Danaus plexippus* reveals variability in dosage compensation across tissues. *Commun Biol* 4:791.
- von Grotthuss M, Ashburner M, Ranz JM. 2010. Fragile regions and not functional constraints predominate in shaping gene organization in the genus *Drosophila*. *Genome research* 20:1084-1096.
- Yanai I, Benjamin H, Shmoish M, Chalifa-Caspi V, Shklar M, Ophir R, Bar-Even A, Horn-Saban S, Safran M, Domany E, et al. 2005. Genome-wide midrange transcription profiles reveal expression level relationships in human tissue specification. *Bioinformatics (Oxford, England)* 21:650-659.

Appendix B

Referee: 2

Comments to the Author(s).

Unfortunately, neither the new draft nor the response to my comments changes my evaluation of the manuscript substantially. I do not understand why the authors don't think that conservation between more deeply-diverged animals diminishes the significance of their finding of conservation between more recently-diverged animals. I do not see that any important general statements about biology hinge on the rate of rearrangements specifically in insects. Therefore, I unfortunately still do not support publication in Proc B.

We again must respectfully dissent from this reviewer. Relative to the first assertion, we surely find interesting that there is conservation between more deeply-diverged animals but we do not think that such observation diminishes the relevance of our finding in insects for the very simple reason that the inferences made from comparative analysis based on deeply-diverged animals tend to involve genomes that are much more static than insect genomes in terms of chromosome repatterning. In other words, conservation of gene organization among deeply-diverged animals might often reflect phylogenetic inertia, not always representing a strong test for alternative explanations (e.g. the presence of regulatory-based constraints). Insects, because of their outstanding properties (e.g. superdynamic genomes and short generation time), do truly pose a much more serious test to the interpretation of signatures of macro- and micro-synteny conservation.

Relative to the second assertion, we believe that we provide in our paper multiple findings with relevant implications for a better understanding of the evolution and analyses of insect genomes. In some cases, such findings are linked to rates of rearrangement, and in some other cases they are not. We would have appreciated specific details explaining why such findings are -in the opinion of this reviewer- not relevant enough.

Lastly, we want to clarify that we do not have any problem in acknowledging that extending our analytical design to other deuterostomes and protostomes will be extremely useful in the years to come to truly understand the limits and magnitude of animal genome evolution at the structural level as well as to reconstruct faithfully ancestral genomes at different phylogenetic scales. According to this, we have slightly modified a few sentences in the second paragraph of the Discussion section.

Referee: 3

Comments to the Author(s).

This is an interesting paper that tries to begin to unravel the nature of the fast genome rearrangement in the insect genomes. The authors have responded to the previous reviewers well and I would thus recommend this paper for publication.

We truly appreciate the encouraging comments from this reviewer.